# Physical Properties and Microstructure of Concrete with Waste Basalt Powder Addition

**DOI:** 10.3390/ma13163503

**Published:** 2020-08-08

**Authors:** Magdalena Dobiszewska, Ahmet Beycioğlu

**Affiliations:** 1Faculty of Civil and Environmental Engineering and Architecture, UTP University of Science and Technology in Bydgoszcz, 85-304 Bydgoszcz, Poland; 2Construction and Building Materials Division, AAT Science and Technology University, 01250 Adana, Turkey; abeycioglu@gmail.com

**Keywords:** concrete, waste basalt powder, compressive strength, water absorption, permeability, microstructure

## Abstract

The natural aggregates are one of the main components in the production of concrete. Although deposits of natural aggregates lie on the earth’s surface or at low depths and belong to common deposits, the shortage of aggregate, especially natural sand, is presently observed in many countries. In such a situation, one is looking for other materials that can be used as a substitute for natural aggregates in mortars and concrete production. This paper presents the results of an experimental investigation carried out to evaluate the potential usage of waste basalt powder in concrete production. For this purpose, the waste basalt powder, which is a by-product of the production of mineral–asphalt mixtures, was substituted with 10%, 20%, and 30% sand replacement. In the experimental program, the workability, compressive strength, water transport properties, and microstructural performances were evaluated. The results showed that the production of concretes that feature a strong internal structure with decreased water transport behavior is possible with waste basalt usage. Furthermore, when waste basalt powder is used as a partial sand replacement, the compressive strength of concretes can be increased up to 25%. According to the microstructural analyses, the presence of basalt powder in concrete mixes is beneficial for cement hydration products, and basalt powder substituted concretes have lower porosity within the interfacial transition zone.

## 1. Introduction

The economic development of the country is reflected, among other things, in the increasing demand for construction projects, and thus the increased production of construction materials. As a result, the demand for materials for construction projects is constantly growing, and the limitations resulting from the need to protect the environment significantly reduce the range of natural resource deposits that can be used for their production. Currently, the most frequently used building materials in the world are cement composites, among which the dominant role is played by structural concrete. One of natural raw materials used in the production of cement composites are natural aggregates.

The production and utilization of aggregates in Europe is about 4 billion tons per year, most of which, i.e., 91% comes from natural deposits [1]. Although deposits of natural aggregates lie on the earth’s surface or at low depths and belong to common deposits, the shortage of aggregate, especially natural sand, is presently observed in many countries. There are also great problems with obtaining new deposits of natural resources despite relatively large geological resources of aggregate deposits. For example, the documented balance resources of natural aggregates in Poland are relatively large. However, based on the much smaller amount of industrial resources, the gravel-sand aggregate reserve is estimated to be only 18 years, and the broken and block stone reserve is estimated to be only 42 years [2]. Due to resource losses, these numbers may be further reduced by 20–30% [3]. The establishment of the Natura 2000 European network of protected areas has resulted in more than 900 documented deposits, including about 500 exploited rock deposits, located in these areas in Poland. The need to comply with the principles of the EU Habitats Directive causes the reduction of approximately 35% of current extraction capacity [4].

In such a situation, one is looking for other materials that can be used as a substitute for natural aggregates in mortars and concrete production. As a partial substitute of fine aggregate, i.e., sand, the blast furnace slag and fly ash can be used. For this, there are many experimental studies available in the literature.

Bilir [5] investigated the effect of non-ground coal bottom ash and non-ground granulated blast furnace slag on the permeability properties of concretes. Rashad [6] reviewed more than 40 studies in the literature in the past 15 years before 2015, and he stated that the blast furnace slag and copper slag can be used as a partiall/full replacement of natural fine aggregate in mortar and concrete. Yuksel and Genc [7] investigated the possibility of using granulated blast-furnace slag, furnace bottom ash, and their combination as fine aggregates in concrete. They used these powder materials without applying any preprocesses such as sieving and grinding, and they reported that concrete strength decreases with increasing replacement ratio with respect to reference concrete.

Bilir et al. [8] reported that the usage of a high volume of fly ash (at the ratio of 60–70%) as fine aggregate without causing significant changes on the properties of mortars is possible. Ravina [9] utilized a large quantity of Class F fly ash in structural concrete as a partial fine sand replacement and found that the fly ash has a good effect on the compressive strength, particularly at a later age. Besides, the author mentioned that the maximum penetration depth of water under the pressure of fly ash substituted mixtures is smaller than that of the reference mix. Siddique [10] substituted the class F fly ash with 10%, 20%, 30%, 40%, and 50% partial sand replacement, and he determined the compressive strength, splitting tensile strength, flexural strength, and the modulus of elasticity after 7, 14, 28, 56, 91, and 365 days of curing. According to the findings, the author mentioned that the C class fly ash can be effectively used in structural concrete.

The rock dust, which is a by-product obtained from the production process of crushed-stone aggregates, is another well-known powder material to be used in concrete. During rock extraction and mechanical processing, and due to their sorting, large quantities of waste material are produced in the form of rock dust. Similar dust waste is produced during the drying process of the aggregate used for the production of mineral–asphalt mixtures and at stonemason facilities. The storage of this type of fine material poses serious environmental problems. Dust released to the atmosphere significantly contributes to the accumulation and harmful dispersion of fine solids in air, water, and soil [11,12].

The chemical and mineral composition of dusts is the same as of the bedrock from which they originate. This makes them suitable for the production of cement mortars and concretes as a partial substitute of fine aggregate or even cement. Not only will it reduce the cost of construction production, it will also make the management of this waste more efficient. Such waste utilization is consistent with the principle of sustainable development that assumes an efficient management of non-renewable (consumable) natural resources and their replacement with recycled waste substitutes.

Most authors believe that the introduction of rock dust as a replacement for the part of sand contributes to the improvement of the mechanical properties of mortars and cement concretes (Figure 1 and Figure 2) as well as durability. As early as 1976, Soroka and Stern [13] noted the beneficial effect of fillers on the mechanical properties of cement mortars. Strength increases with the dust share in sand mass and with the increase of its fineness. The same conclusions have been drawn by many authors who have studied mortars and concretes with the following additives as replacement for sand lime dust [14,15,16,17,18], marble dust [15,19,20,21], granite dust [12,22,23,24,25,26], and basalt dust [14,27].

Abdelaziz et al. [14] studied the use of two different quarry dusts as cement and fine aggregate replacement materials and experimentally demonstrated the usability of quarry dust as fine aggregate replacement. It was concluded that the strength properties of blended cement mortars with quarry dusts are higher than those of the control mortars. Binici et al. [15] used marble dusts and limestone dusts instead of fine sand aggregate. The authors stated that higher abrasion resistant concrete can be produced by using this waste instead of sand. Furthermore, the authors reported that the sulfate resistance also increased in concretes containing this waste. The study of Çelik and Marar [16] revealed that the use of breaker powder in concrete has been worked on for a long time. The authors noticed that the addition of dust improved the compressive strength and abrasion resistance, while reducing the absorption and permeability of concrete. Eren and Marar [17] found a reduction in water permeability with the increasing replacement level of the crusher dust replacement in fine aggregate. Topçu and Uğurlu [18] used the mineral filler as a replacement for sand, and they investigated the effect of applying different amounts of mineral filler on concrete. They found that the addition of 7–10% of mineral filler to fine aggregate (0–2 mm) considerably improved the mechanical properties of concrete and decreased its permeability. Aliabdo et al. [19] studied the utilization of waste marble dust in cement and concrete production. They added the marble dust with 0.0%, 5.0%, 7.5%, 10.0%, and 15.0% replacement ratios by weight of cement and sand. They concluded that concrete made of marble dust as a sand replacement indicated better performance compared to cement replacement. Alyamaç and Aydın [20] produced the concrete samples by replacing sand with marble powder at 10%, 20%, 30%, 40%, 50%, and 90% by volume. They concluded that using up to 40% marble powder in concrete had a positive effect on the mechanical properties of concrete, including abrasion resistance, and it also reduced water absorption. Corinaldesi et al. [21] used a by-product of marble sawing and shaping in concrete production and it was found that 10% substitution of sand with marble powder provided maximum compressive strength. Arivumangai and Felixkala [22] studied the usage of granite powder with 0, 25%, and 50% sand replacement. They mentioned that the concrete with increased performance in compressive strength as well as in the durability aspect can be produced. Bonavetti and Irassar [24] used the stone dusts of quartz, granite, and limestone in percentages ranging from 0 to 20% as a replacement for equal weights of sand. They found an improvement in mortar strength at early ages, and no detrimental effects were observed at later ages. Reddy et al. [25] and Divakar et al. [26] studied waste granite powder and concluded that locally available granite is a useful partial substitution material for concrete. This material can improve the compressive, tensile, and flexure strengths of concrete.

The improvement of properties of cement composites with rock dust additives is primarily related to the filler role of rock dust. The operation mechanism of chemically inert dust neither depends on the type of rock material the dust originates from nor its chemical composition [31,32]. Therefore, the dust grain size of rock dust is much more important in these cases. Fine dust material acts as an inert filler, which contributes to better filling of the intergranular free space in a composite. This results in a compact cement matrix structure with lower porosity and therefore greater strength and durability [24,33,34]. Apart from the dominant filler effect, the heterogeneous nucleation of C–S–H on rock dust grains plays a secondary role in shaping the microstructure of the hardened cement slurry, which leads to the increased content of hydrated calcium silicates [35,36]. This contributes to the additional sealing of the cement matrix and thus to its strength increase.

The use of waste materials in the production of cement composites, the properties and evaluation criteria of which have not been specified in the standards, requires a comprehensive analysis and evaluation of the properties of mortars and concretes produced with their use. Despite the growing interest in the use of various rock dust types in concrete production, many problems still remain unexplained. There is no literature related to basalt dust utilization as a partial sand substitute in concrete. The scientific goal of the research presented in the paper was to determine the effect of the basalt dust additive, partially replacing sand, on the properties of concrete. The influence of basalt dust on the technological properties of concrete mixtures, including the compressive strength, water absorption, permeability, and microstructure of hardened concrete with basalt dust additive were analyzed. The results of research on the concrete mechanical properties presented by the authors in this paper are consistent with those of other scientists who analyzed the influence of rock dusts of different mineral origin (lime, marble, and granite dusts) on concrete properties. Moreover, the presented analyses concerning the influence of basalt dust on the porosity and microstructure of the cement matrix supplement the knowledge deficiency, especially with regard to basalt dusts used in concrete as a sand substitute.

## 2. Materials and Methods

Waste basalt powder generated in the production process of asphalt mixtures with basalt aggregate was used. During the drying of the mineral aggregate, exhaust fumes leave the dryer with various powder particles. A coarser fraction of waste powder is collected in a special separator, while a very fine fraction is retained in a dryer filter. Asphalt mixture production leads to the formation of rock powder in an amount of about 5% of the aggregate mass used to produce the asphalt mixture. This very fine material is treated as a waste.

The chemical composition of the basalt powder used in this study is presented in Table 1. Figure 3 presents the particle size distribution of the basalt powder, which is similar to OPC (ordinary Portland cement). The range of the basalt powder particle diameters is 0.5 to 200 µm, and the average particle size is 20 µm in diameter. The basalt powder specific gravity is 2.99, and the specific surface area determined by the Blaine method is 3500 cm^2^/g.

Basalt powder particles have a rough surface and an angular shape (the scanning electron microscope SEM image (Scanning Electron Microscope (SEM) Quanta 250 FEG by FEI (Hilsboro, OR, USA), equipped with the system of chemical content analysis based on the energy dispersion of the X-ray using Energy Dispersive X-ray Spectroscopy (EDS, Panalytical, Almelo, The Netherlands) by EDAX) is presented in Figure 4). The mineralogical composition of the basalt powder was determined based on the XRD diffractogram (X-ray diffraction (XRD) method using a X’pert MPD X-ray diffractometer (Panalytical, Almelo, The Netherlands) with a goniometer PW 3020, Cu lamp and a graphite monochromator. Diffraction patterns were recorded by step scanning from 5 to 65, with a step size of 0.02°. HighScore Pro software version 4.1 (Panalytical, Almelo, The Netherlands) was used to process diffraction data. The identification of mineral phases was based on the PCPDFWIN ver. 1.30 formalized by JCPDS-ICDD (ICDD, Newtown, CT, USA)) (Figure 5). Plagioclase rich in anorthite particles (Ca-Plagioclase) dominates in the mineral composition of basalt powder, as well as pyroxene and amphibole. There is a small amount of illite, which is probably the effect of plagioclases weathering.

The concrete mixtures were prepared with the use of ordinary Portland cement CEM I 42.5R (Lafarge Cement Plant Kujawy in Bielawy, Poland). Table 2 presents the chemical and mineral composition of the OPC. The particle size distribution of cement is shown in Figure 3. The ordinary Portland cement specific gravity is 3.13, and the specific surface area determined by the Blaine method is 3500 cm^2^/g. As a coarse aggregate (CA) (Pędzewo, Zławieś Wielka, Poland), the gravel of the group of fractions 2/16 was used, and as a fine aggregate (FA) (Pędzewo, Zławieś Wielka, Poland), river sand (Pędzewo, Zławieś Wielka, Poland) was used. In order to obtain the desired workability of concrete mixtures, the high-range water-reducing (HRWR) admixture was added.

To analyze the effect of basalt powder on the properties and microstructure of concrete, four concrete mixes were prepared. The reference concrete, i.e., concrete without waste basalt powder, was named C0, and the concretes with different amount of basalt powder replacing 10%, 20%, and 30% of the sand by mass were named C10, C20, and C30, respectively. The comparison between the properties and microstructure of reference concrete without basalt powder and concretes containing different amounts of waste basalt powder was made. The composition of the concrete mixtures is presented in Table 3. The water/cement ratio was maintained constant at 0.4.

To analyze the technological properties of concrete mixtures, the concrete slump test according to European Standard EN-12350-2:2011 was performed. Maintaining the same consistency and similar workability level of each concrete mixture was assumed, therefore with increasing basalt powder content, the high-range water-reducing admixture amount was increased.

To assess the influence of different amounts of basalt powder on the physical properties of concrete, the compressive strength, permeability, and water absorption were analyzed. Compressive strength and permeability were conducted according to European Standards EN-12390-3:2011 and EN-12390-8:2011. The compression tests were performed using a computer-controlled test machine of 3000 kN capacity. The hardness value of the compressive test machine loading heads is 550 HV 30 (HRC 53), which conforms to the EN12390-4. For tests, the loading rate was selected constant as 0.5 MPa/s. The permeability of concrete was determined with a measuring device designed to determine the depth of water penetration in hardened concrete specimens under pressure. Research was conducted in accordance with the procedure set out in EN 12390-8. The measuring device enables the transfer of water pressure to the test area and its current indications. Water absorption was determined based on the relative mass loss of samples dried up to constant mass compared with samples fully saturated with water. Cube specimens 100 mm × 100 mm × 100 mm and 150 mm × 150 mm × 150 mm (for permeability determination) were prepared for each concrete, i.e., the reference concrete and concretes with waste basalt powder. A laboratory mixer was used to prepare concrete mixtures. After placing in the molds, concrete specimens were consolidated in two layers by using mechanical vibration. Until demolding, all specimens were kept covered in a chamber at controlled temperature 20 ± 2 °C for 24 h. After demolding, specimens were stored in water until testing. The testing of compressive strength was conducted at 7, 14, 28, 90, 180 and 360 days, the testing of permeability was conducted at 28 days, and the testing of water absorption was conducted at 28, 180, and 360 days.

In order to determine the effect of basalt powder on the cement matrix microstructure, the scanning electron microscope (SEM) observation and energy-dispersive spectroscopy (EDS) analysis were performed. SEM analyses were performed with Quanta 250 FEG by FEI (Hilsboro, OR, USA) instrument, equipped with the system of chemical content analysis based on the energy dispersion of the X-ray using Energy Dispersive X-ray Spectroscopy (EDS, Panalytical, Almelo, The Netherlands) by EDAX.). Fractured samples were specially prepared. The phases presented in the interfacial transition zone and bulk hydrated cement paste were identified based on EDS analysis.

## 3. Results and Discussion

The partial replacement of sand with basalt dust leads to a change in two parameters of the analyzed concrete, i.e., the particle size distribution of the aggregate and rock material (quartz on basalt). The introduction of basalt dust into the concrete mix in exchange for the corresponding sand dust fraction allows maintaining the same screening curve of the aggregate mix and analyzing the impact of the type of rock material on the properties of concrete. However, in such a situation, it is difficult to talk about the physical effect of the additive, i.e., the effect of the filler, because the sand dust fraction is replaced with the same dust fraction. The addition of mineral dust as a substitute exclusively for the fine aggregate dust fraction will affect the properties of concrete when this additive has a clear activity in relation to the solution in the pores of the hardened cement paste. Conducting research with the substitution of sand with basalt dust allowed, first of all, to analyze the influence of aggregate mixture sealing on the properties of mortars and concretes.

### 3.1. Properties of Concrete Mixtures

The change in concrete mix consistency, measured with a slump cone test, when the level of sand substitution with basalt dust at constant and variable fluidizing admixture content increases is presented in Figure 6. Incorporating dust in the concrete mix leads to a change toward a less liquid mix consistency, as well as to a deterioration of workability.

The results of the conducted tests show that a constant content of fluidizing admixture causes the slump cone test to vary from 140 mm, in a mixture without basalt dust, to 40 mm when 30% of sand is replaced with dust. This denotes the change of the concrete mix fluidity from an S3 to S1 consistency class according to European Standard EN-206:2016. The decreased workability of concrete mix with basalt powder addition is attributed to the greater compactness of the concrete mixture. As a result of the significant sorptivity of basalt powder, water is absorbed by its particles when mixing, which leads to a workability reduction of the concrete mixture [37]. Basalt dust features a large specific surface area, which needs more water to obtain proper consistency than in the case of sand. Therefore, when the basalt dust share in the sand mass increases, the concrete mixture fluidity significantly decreases.

When modifying the concrete composition, it was assumed that the same consistency of concrete mixture, characterized by the slump cone (or Abrams cone) test, was maintained within 140 ± 10 mm. Therefore, as sand was gradually replaced with basalt dust, it was necessary to increase the amount of high-range water-reducing (HRWR) admixture. At 30% basalt dust content, the fluidizing admixture content has more than doubled compared to the reference mixture.

The results of studies on the influence of basalt dust on the properties of concrete mixtures with constant consistency with an increasing amount of basalt dust replacing sand are presented in Table 4. As the basalt dust content increases, the air content decreases significantly from 4.1% in the case of the reference mixture to 2.5% in a 30% added dust mixture. The gradual substitution of sand with basalt dust leads to the sealing of aggregate composition, which results in lower air content in the concrete mix and thus a lower porosity of hardened concrete. Along with the increased basalt dust content in the sand mass, the increase in concrete mixture density is observed. The basalt dust density is 2.99 g/cm^3^ and is higher than sand density (2.65 g/cm^3^), which is the reason why the density of the concrete mixture increases with the dust share increase in the sand mass.

The introduction of dust material into the concrete mix reduces bleeding [38]. The content of very small aggregate grains with a diameter of less than 150 μm reduces water bleeding from the concrete mix mainly due to the lower sedimentation rate of fine grains [39].

### 3.2. Compressive Strength

Obtained results clearly indicate the positive influence of basalt dust as a sand replacement on the concrete strength (Figure 7). With the increase of basalt dust content, the compressive strength of concrete increased in each curing time.

The greatest strength increase in relation to the reference concrete, i.e., 19%, 24%, 25%, and 23% was reported respectively after 14, 28, 90, and 180 days of hardening, when replacing 30% of sand with basalt dust. Then, in the case of early 7-day strength, the highest 13% increase was obtained in concrete with 20% basalt dust additive. Concrete strength with basalt dust after 360 days of hardening does not significantly differ from that of the reference concrete. It is worth mentioning that concrete with 10% basalt dust additive gained 28-day reference concrete strength just after 7 days and 90-day reference concrete strength after only 14 days.

The beneficial effect of basalt dust on the mechanical properties of concrete is connected with the physical effect of this additive, i.e., with the filler effect. Basalt dust grains fill empty spaces (pores) between sand grains and coarser aggregate. This results in a higher tightness of the aggregate composition, a more compact cement matrix microstructure, and thus lower porosity and higher material strength [24,33]. From the grain size distribution of basalt dust and cement shown in Figure 3, it can be seen that the granularity distribution of both materials is similar. However, basalt dust is of the bimodal grain size type. The largest volume is occupied by grains with a diameter of approximately 20 µm. However, on the basalt dust grain curve, there is another, smaller maximum corresponding to a grain diameter of 1 µm. These very small dust particles were most likely located among slightly larger cement grains, which sealed the microstructure of the cement matrix as well as contributed to improvements in the mechanical properties of mortars and concretes.

The introduction of basalt dust has contributed to an increased level of active centers where it is possible to crystallize hydration products, especially the C–S–H phase, which also has an influence on the increased mortar strength. Cement grains, electrostatically charged differently, tend to attract one another in water slurry, resulting in aggregate creation (flocculation). In this case, water is not able to freely penetrate all the space between cement grains, and not all cement particles are effectively used in the hydration process. Microfiller introduction causes a greater dispersion of cement grains, which contributes to the accelerated hydration of clinker phases and therefore a faster strength increase [12,33].

### 3.3. Water Absorption

Basalt dust additive has little effect on the water absorption of concrete. The mass absorption of all tested concretes did not exceed the value of 4.5% and decreased with hardening time (Figure 8). After 28 days of hardening, the absorption of all tested concretes was identical. In longer periods, i.e., after 180 and 360 days of hardening, the absorption began to decrease slightly with the increase of basalt dust content. Water absorption decrease with increasing hardening time is attributed to the gradual completing hydration, which leads to the better compactness of cement matrix and the refinement of the pore structure [36,39].

It is worth mentioning that the concrete absorption was determined without evaluating the actual volume of water-saturated material, which may significantly affect the test results. This specific test method uses the mass of water absorbed by the concrete to refer to the mass of the whole dried sample and not only to the mass of the volume of the water-saturated material. It may have a particular effect when determining the absorption of concrete after 28 days of hardening. A longer sample hardening period in wet conditions means a larger volume of water-saturated samples. Therefore, it can be considered that a more reliable assessment of the impact of basalt dust on the concrete absorption can be performed after a longer hardening period, i.e., after 180 and 360 days of concrete cure.

### 3.4. Permeability

The research showed that the penetration depth of pressurized water in all analyzed concretes is very low and is between 10 and 17 mm (Table 5). Here, 50 mm of water penetration allows defining concrete as impermeable. However, concrete in which the depth of water penetration under pressure does not exceed 30 mm meets the requirements of impermeable concrete in corrosive conditions [37].

Although all concretes feature low permeability, the water penetration depth in concrete with basalt dust additive is slightly lower in comparison with the reference concrete. The pore system has a significant influence on the permeability of concrete. The reduction of permeability is strongly influenced by the pore system. The share and structure of continuous capillary pores is of the greatest importance here [37]. The interruption of capillary pores reduces the penetration and movement of water in the concrete, thus reducing its permeability. Very small basalt dust particles most likely blocked the continuous capillary pores, which undoubtedly reduced the concrete permeability.

### 3.5. Microstructure

In order to analyze the microstructure, porosity tests of cement mortar samples were performed. Basalt dust was a sand substitute in the amount of 10% and 20%. It was found that as the basalt dust share increases, the total pore volume and porosity decreases. Basalt dust additive reduces mortar porosity in the capillary pore range. On the other hand, the share of fine pores below 50 nm in diameter and of gel pores below 10 nm is significantly increased (Figure 9). This has a positive effect on mortar microstructure sealing, which translates into increased strength and durability. A higher share of fine pores in the total material porosity means greater strength at the given porosity [37,40,41,42,43]. This is in line with the results of strength tests of concrete with basalt dust additive as a sand replacement, which show that mortar strength increases with the increase in basalt dust content. The basalt dust used in the study features a bimodal grain size distribution with a small maximum around 1 µm in diameter (Figure 3). These very small particles of basalt dust caused the cement matrix microstructure to be sealed, resulting in lower porosity. The heterogeneous C–S–H nucleation on dust particles, which results in a higher C–S–H content and thus a lower porosity of the basalt dust slurry that replaced sand, is not to be underestimated. The reduction of porosity across capillary pores undoubtedly reduced the absorption and permeability of concrete with basalt dust additives.

From SEM observation and EDS analysis, it was found that products of hydration, the C–S–H phase in particular, crystallize on basalt dust particles. Very fine particles of basalt dust act as crystallization centers and provide additional areas where C–S–H nuclei can settle (Figure 10 and Figure 11).

In cement paste without basalt dust additive, large hexagonal portlandite plates were identified (Figure 12), while these were not found during the research of paste microstructure with basalt dust addition. In many places, large portlandite crystals in the interfacial transition zone between aggregate and paste in concrete without basalt dust were also found (Figure 13), which was not noticed in the case of concrete with dust additive (Figure 14). However, the results of research on the calcium hydroxide content in cement paste did not confirm that the basalt dust additive reduces its content. Therefore, it can be assumed that basalt dust particles finer than cement grains sealed the cement paste microstructure, as a result of which there were no large pores giving space for the formation of larger portlandite crystals. Much smaller, evenly dispersed portlandite crystals started to form in the cement matrix, and they were more difficult to identify in the images of the SEM microstructure.

The presented SEM image of a concrete fracture with basalt dust additive (Figure 14) contains a visible compact interfacial transition zone that consists mainly of the C–S–H phase, which adheres tightly to the aggregate grain surface. Within the interfacial transition zone, very small basalt dust grains are visible, which tightly adhere to the mass of hardened cement paste (points 1 and 3 in Figure 14 and Figure 15). The lack of large portlandite crystals indicates the lower porosity of the interfacial transition zone in concrete with basalt dust additive, which, as we know, indicates that this zone has been reinforced.

## 4. Conclusions

This study discusses an experimental program carried out to investigate the effects of basalt powder replacement on the physical, mechanical, and microstructural properties of concretes. The following conclusions can be drawn based on the results of this study:Basalt powder affected the workability of fresh concretes. When mixing water is kept constant in concrete mixtures, the workability of concrete mixes decreased with the increase in substitution ratios of basalt powder. When the basalt powder replacement rate was increased to 30%, a 71% decrease in consistency occurred (slump value decreased from 140 to 40 mm). For the production of basalt powder substituted concrete with constant consistency, HRWR should be increased based on the increasing rate of basalt powder substitution. Thus, the consistency of concretes can be maintained by increasing HRWR based on the increase of basalt powder. This increase in water demand can be attributed to the relatively greater surface area of basalt powder.The amount of air in the reference mixture, which is 4.1%, decreased up to 2.5% with 30% basalt powder addition. This indicates that the use of basalt powder reduces the void rate in the concrete. In addition, concrete densities also increased inversely proportional to the amount of air and supported the improvement in the internal concrete structure. This result can be related to the higher density of basalt powder compared to the sand used in the mixture. This means that the concrete internal structure has become more compact and the impermeability has increased with basalt powder substitution, and thus a better concrete internal structure has been obtained.The compressive strength of concretes increased in all mixes based on the concrete curing age. In addition to this well-known result, the substitution of basalt powder instead of sand increased the compressive strength of all substitution rates and curing periods. In concrete mixes containing basalt powder, the increase in compressive strength has been achieved up to 25%. The increase in compressive strength, which is an expected result due to the decrease in the amount of concrete air and the increase in density, can be explained by the fact that the internal structure of the concrete becomes more compact as a result of basalt powder usage.Although there is no significant relationship between water absorption and basalt substitution rate, very clear information was obtained from the pressurized water penetration depth data. As it is well-known, the pore system has a significant influence on the permeability of concrete. According to the achieved findings, it can be concluded that the relatively small particles of basalt dust blocked the continuous capillary pores and thus reduced the concrete permeability.Microstructural analyses showed that the presence of basalt powder in concrete mixes is beneficial for cement hydration products. According to the SEM observation and EDX analysis, very fine particles of basalt dust acted as crystallization centers and provided additional areas where C–S–H nuclei can settle. It was observed that the basalt powder well adhered to the hydrated cement paste, and the portlandite crystals did not appear in large quantities between the basalt powder and the C–S–H gels. Contrary to reference mixes, the lack of large portlandite crystals was observed in the microstructures of basalt powder-substituted concrete. This observation indicates that basalt powder-substituted concretes have lower porosity through the interfacial transition zone. This is the result of the reinforcement effect of basalt powder on ITZ (interfacial transition zone).

As a result, the experimental study showed that in addition to improving some physical, mechanical, and microstructural performances of concretes and reducing the usage of natural raw materials, using basalt dust leads to the consumption of this industrial waste, thus providing a twofold benefit.

This study may be extended with new experimental studies related to concrete durability and the steel–concrete bond performance of basalt dust-substituted concretes to better understand the potential of basalt powder usage in concrete production.

## Figures and Tables

**Figure 1 materials-13-03503-f001:**
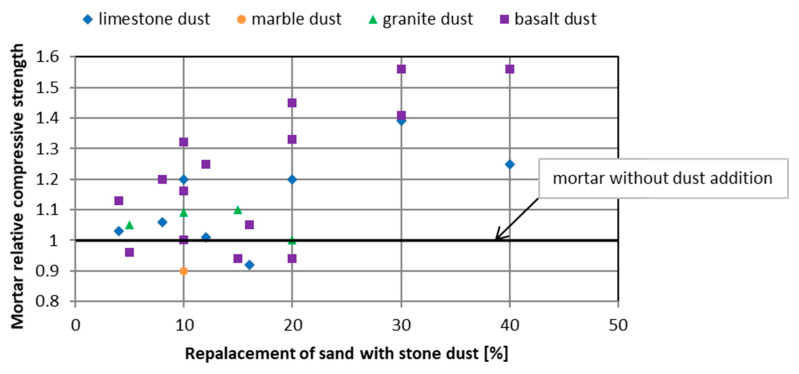
The influence of rock dust on the compressive strength of cement mortars after 28 days of hardening [14,21,24,27,28].

**Figure 2 materials-13-03503-f002:**
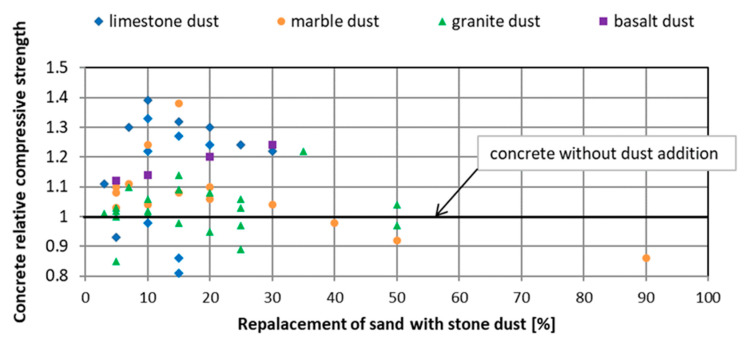
The influence of rock dust on the compressive strength of concrete after 28 days of hardening [12,15,16,17,18,19,20,22,23,25,26,29,30].

**Figure 3 materials-13-03503-f003:**
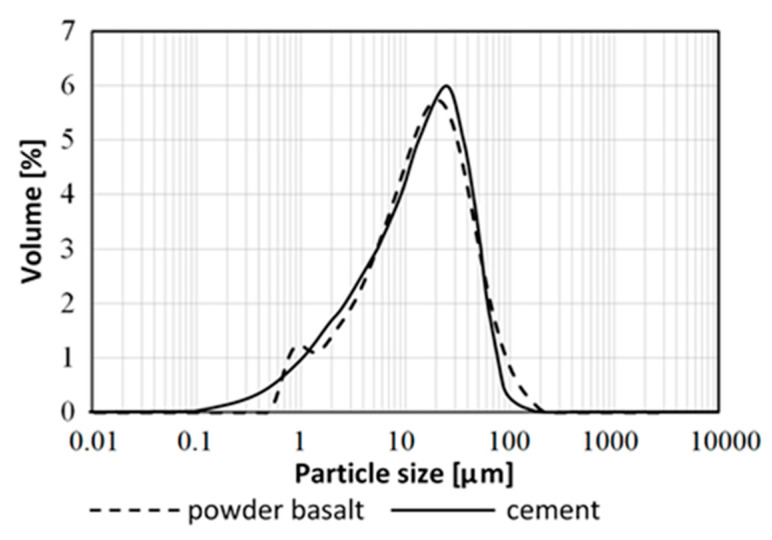
Particle size distribution of basalt powder and Portland cement.

**Figure 4 materials-13-03503-f004:**
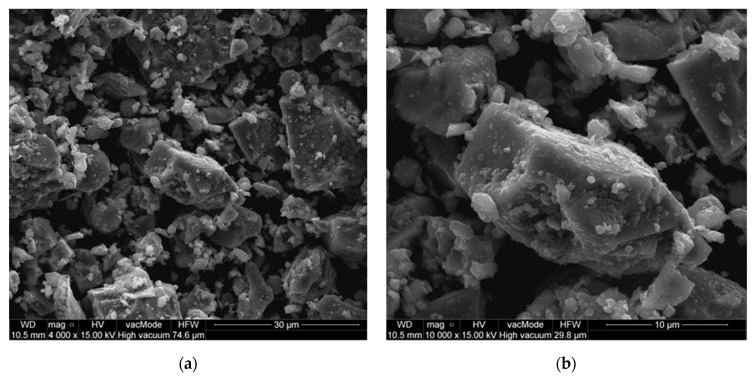
Scanning electron microscope image of basalt powder: (**a**) mag. ×4000; (**b**) mag. ×10000.

**Figure 5 materials-13-03503-f005:**
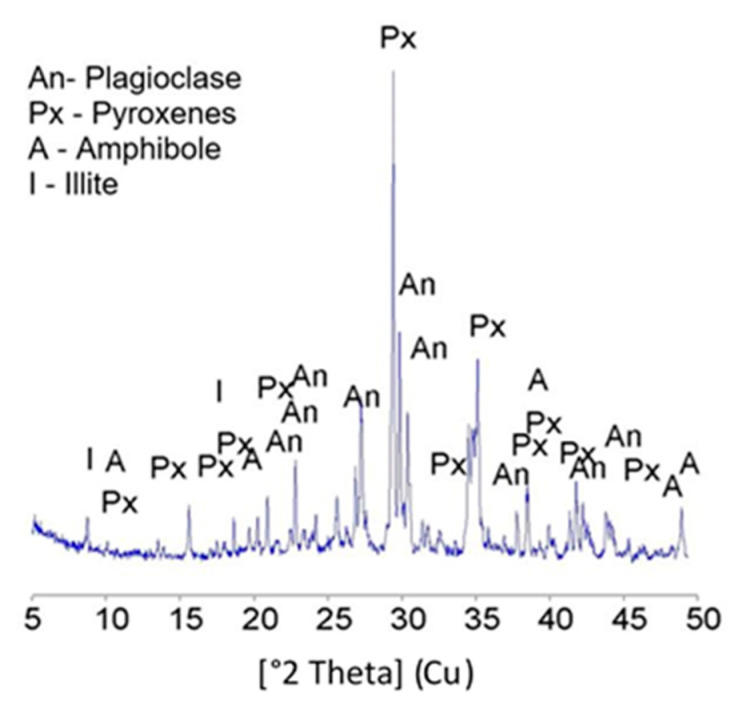
XRD diffractogram of the basalt powder.

**Figure 6 materials-13-03503-f006:**
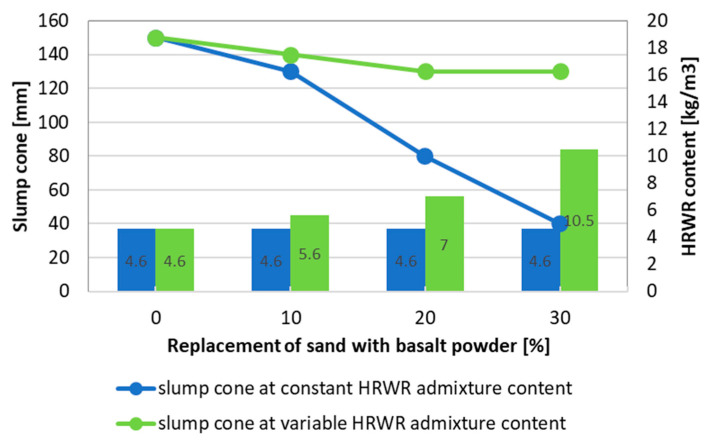
Change of the value of concrete mix slump cone test as a function of basalt dust content at constant and variable HRWR admixture content.

**Figure 7 materials-13-03503-f007:**
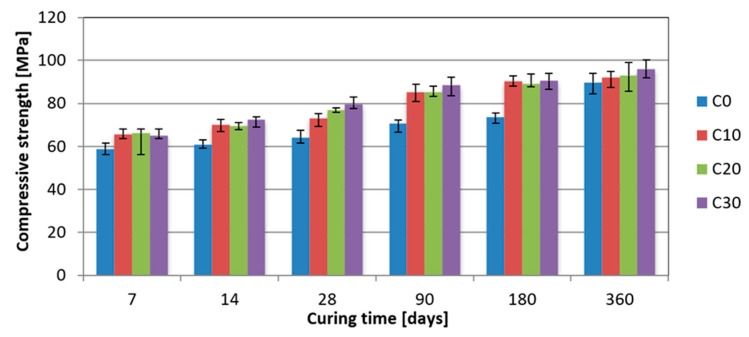
Compressive strength of concrete as a function of time and content of basalt dust as sand replacement (C0–reference concrete, C10–C30–concrete with additive of basalt dust replacing 10–30% of sand).

**Figure 8 materials-13-03503-f008:**
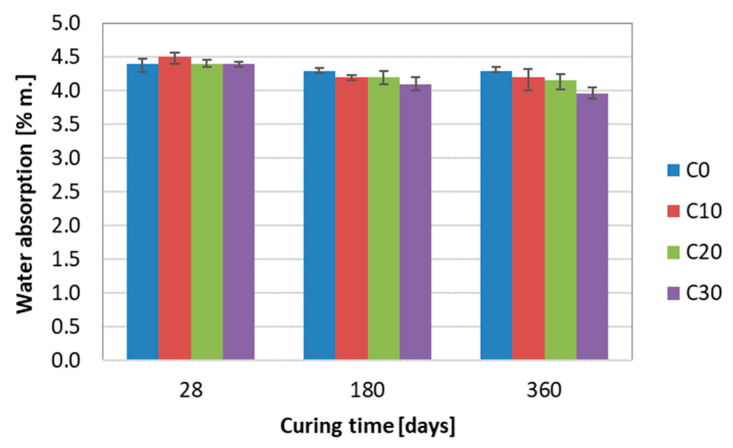
Water absorption of concrete as a function of basalt dust content and hardening time.

**Figure 9 materials-13-03503-f009:**
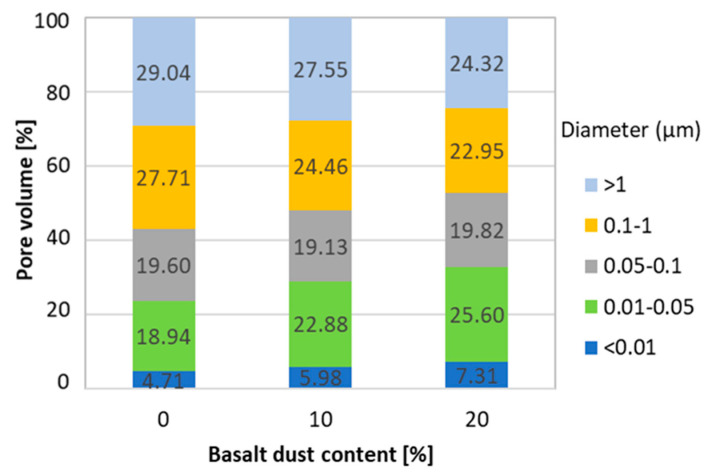
Share of pores of a certain diameter in the total pore volume in cement mortar with basalt dust additive [44].

**Figure 10 materials-13-03503-f010:**
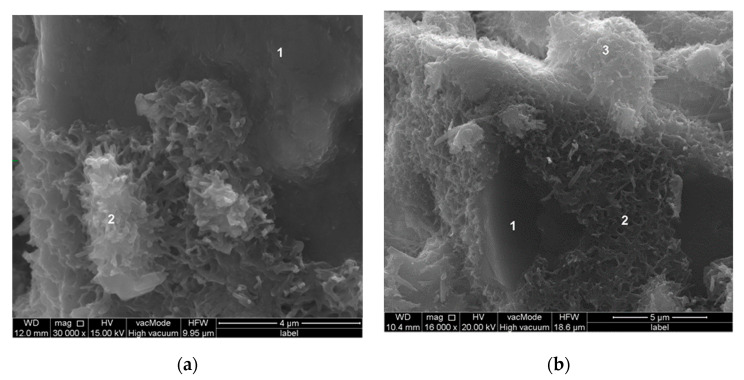
Microstructure of paste with basalt dust: (**a**) after 2 h of hydration; (**b**) after 5 h of hydration; point 1—basalt dust grain, points 2 and 3—C–S–H phase.

**Figure 11 materials-13-03503-f011:**
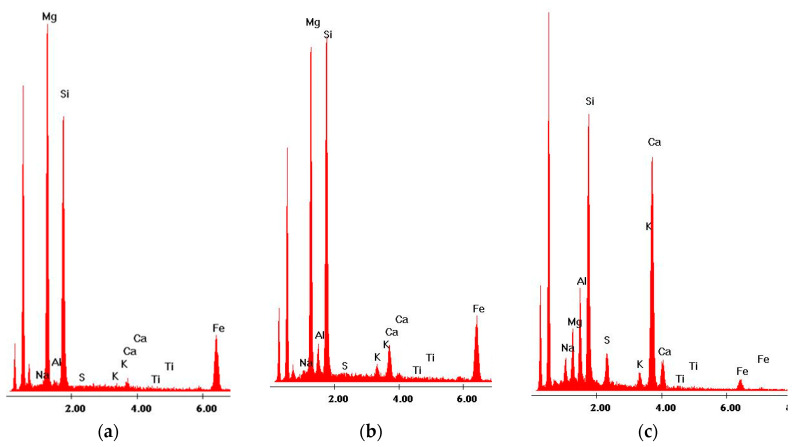
X-ray analysis in the micro-areas indicated in Figure 10: (**a**) olivine crystal—point 1, (**b**) and (**c**) thin layer of C–S–H phase on olivine grain—points 2 and 3.

**Figure 12 materials-13-03503-f012:**
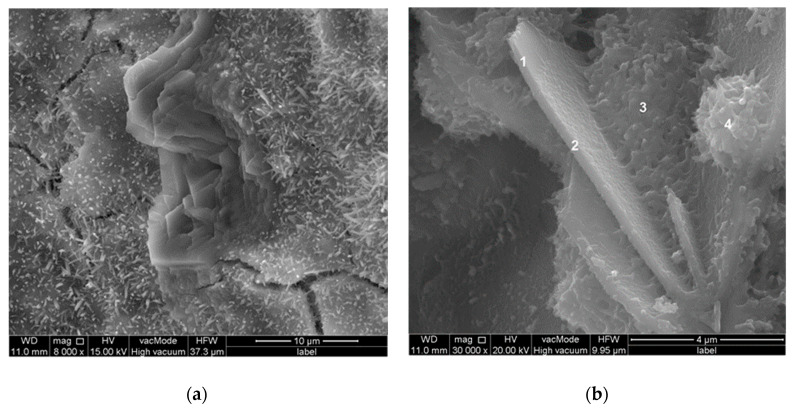
Microstructure of cement paste without basalt dust; (**a**) After 2 h of hydration; (**b**) After 5 h of hydration. Large hexagonal portlandite plates are visible: points 1 and 2—portlandite, points 3 and 4—C–S–H phase.

**Figure 13 materials-13-03503-f013:**
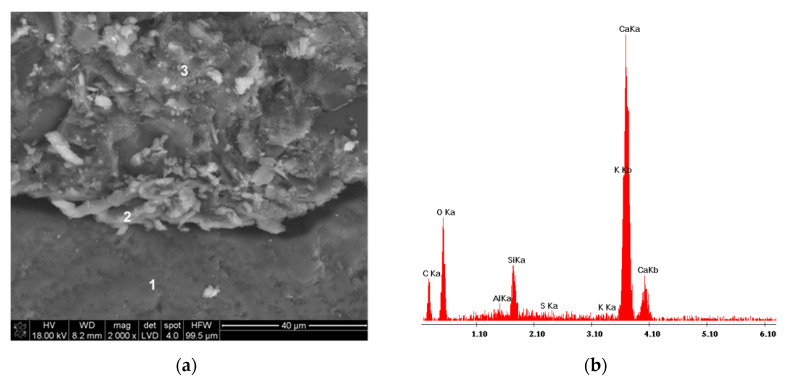
Microstructure of interfacial transition zone: (**a**) Concrete without basalt dust after 28 days of hydration: point 1—aggregate grain, point 2—portlandite, point 3–C–S–H phase; (**b**) Analysis in the micro-area defined in point 2.

**Figure 14 materials-13-03503-f014:**
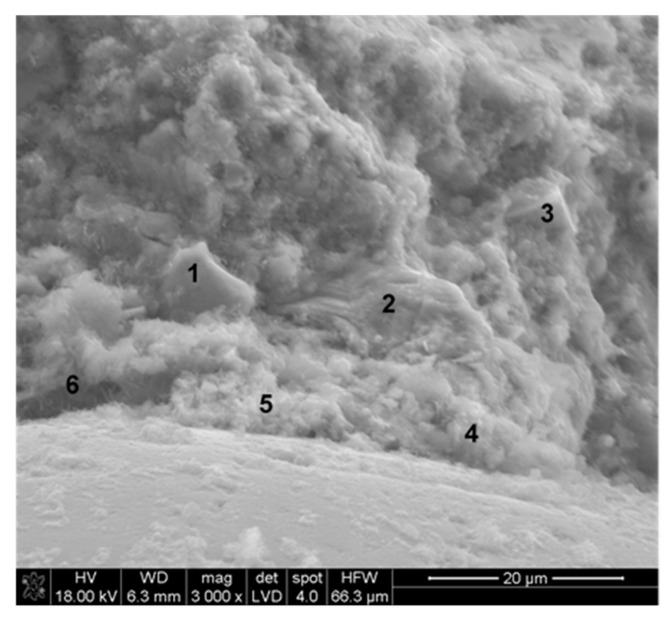
Microstructure of interfacial transition zone: concrete with 10% basalt dust addition after 28 days of hydration: point 1 and 3—basalt dust grain, point 2 and points 4–6—C–S–H phase.

**Figure 15 materials-13-03503-f015:**
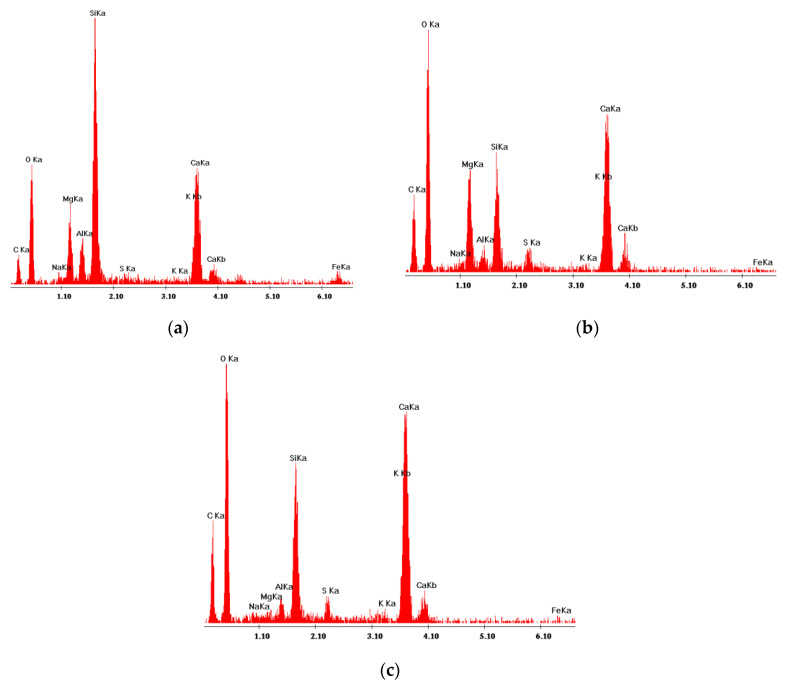
X-ray analysis in micro-areas indicated in Figure 14: (**a**) and (**b**) Basalt dust grains—points 1 and 3; (**c**) C–S–H phase—point 5. C–S–H was also found in points 2, 4, and 6, the X-ray analyses of which were not included.

**Table 1 materials-13-03503-t001:** Chemical composition of basalt powder.

Chemical Composition (%)	
SiO_2_	42.61
Al_2_O_3_	12.90
Fe_2_O_3_	14.05
CaO	13.00
MgO	7.82
SO_3_	0.07
K_2_O	1.15
Na_2_O	1.76
Cl^−^	0.10
P_2_O_5_	1.80
MnO	0.25

**Table 2 materials-13-03503-t002:** Chemical and mineral composition of cement.

Chemical Composition (%)		Mineral Composition (%)	
SiO_2_	19.39	C_3_S	59.7
Al_2_O_3_	4.67		
Fe_2_O_3_	3.34		
CaO	63.17	C_2_S	12.4
MgO	1.24		
SO_3_	2.95		
K_2_O	0.62	C_3_A	2.8
Na_2_O	0.17		
Cl^−^	0.07	C_4_AF	11.8
P_2_O_5_	0.12		

**Table 3 materials-13-03503-t003:** Concrete mixture proportions. C10, C20, and C30: concretes with different amount of basalt powder replacing 10%, 20%, and 30% of the sand by mass, CA: coarse aggregate, HRWR: high-range water-reducing.

Concrete	Cement (kg/m^3^)	Water (kg/m^3^)	Basalt Powder (kg/m^3^)	Fine Agg. (kg/m^3^)	CA		HRWR Admix. (kg/m^3^)
					2/8 mm (kg/m^3^)	8/16 mm (kg/m^3^)	
C0	350	140	0	676	512	640	4.6
C10			68	608			5.6
C20			135	541			7.7
C30			203	473			11.6

**Table 4 materials-13-03503-t004:** Properties of concrete mixes with basalt dust additives.

Concrete Mix	Concrete Slump Test (mm)	Air content (% vol.)	Density (kg/m^3^)
M0	150	4.1	2.31
M10	140	3.2	2.41
M20	130	3.0	2.44
M30	130	2.5	2.51

**Table 5 materials-13-03503-t005:** Depth of water penetration in concrete with basalt dust additive.

**Concrete**	C0	C10	C20	C30
**Penetration Depth (mm)**	17	15	10	13

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
