# Peer review of "Physical Properties and Microstructure of Concrete with Waste Basalt Powder Addition"

_materials, 2020, doi:10.3390/ma13163503_

Round 1

Reviewer 1 Report

General comments:

The article presented the importance of recycling and controlling the depletion of natural resources, specifically the recycling of basalt powder. The topic is not new and there is a good amount of literature in area. The state-of-art is not defined well and makes the article a bit weak. The language needs a lot of work in term of grammar and cohesion.

Here are some specific comments:

Line 11: Since the article discusses concrete research, “cement composites” may not be the suitable term here.

Section 1: The introduction is mainly about the compressive strength of concrete. However, it missed the literature about the permeability, absorption, microstructure, and durability of concrete, which are the main part of the article.

Line 70-72: The Authors cited several articles with different additives and replacements. However, in Figure 1, the whole figure is cited from [29]. Does the data in the figure (Figure 1) reproduced from [29] or collected from by Authors from the cited collection of articles?

Line 82-86: This paragraph is not clear. It is better to put more details about the inert materials (iron oxides) and its effect on the final product.

Table 1: What are the side effects of having high amount of Iron Oxide? Are there any negative impact in long-term?

Line 158: Do you mean Figure 3, not Figure 1?

Line 164: Grammar checks are must. For example, in Line 164 “To analysed” and in Line 171: “To analysis”!

Line 208: Slump target 140 ± 10 mm

Section 3: This section should be sub-divided to discuss different tests and results. In addition, the section presented the discussion before showing results. More scientific elaboration is essential during discussion of results.

Line 272-273: It worth to explain more about the relation between dispersion and hydration of cement.

Line 295: concrete cure.

Section 4: it looks more like a summary rather than conclusions. It should be more concise.

Line 408: Authors mentioned “fully justified”. However, there are so many other parameters that are not covered by this article and may have drawbacks on the long-term and durability.

Author Response

Dear Reviewer

Thank you for your comments concerning our manuscript entitled “Physical Properties and Microstructure of Concrete with Waste Basalt Powder Addition”. The comments were all valuable and very helpful for revising and improving our paper, as well as for helping us to better explain the significance of our research. We have studied the comments carefully and have made a variety of revisions that we hope will meet with your approval. The revised portions of the manuscript are highlighted in yellow in the manuscript itself. In addition, the main revisions to the paper are discussed in our individual responses to your comments that follow below:

Line 11: Since the article discusses concrete research, “cement composites” may not be the suitable term here.

Response to the reviewer:

The term “cement composites has been changed into “concrete” (line 11).

Section 1: The introduction is mainly about the compressive strength of concrete. However, it missed the literature about the permeability, absorption, microstructure, and durability of concrete, which are the main part of the article.

Response to the reviewer:

The introduction has been changed according to the reviewer suggestion (line 114-145 highlighted in yellow in the manuscript). We added the review of the literature about other properties of concrete, i.e. permeability, abrasion resistance, durability, and water absorption.

Line 70-72: The Authors cited several articles with different additives and replacements. However, in Figure 1, the whole figure is cited from [29]. Does the data in the figure (Figure 1) reproduced from [29] or collected from by Authors from the cited collection of articles?

Response to the reviewer:

The date in the Figure was collected from the collection of articles. It was presented in the article [29] of one of the authors. It has been changed in the paper, i.e. all cited articles have been added to the references (line 103, 112, 113 highlighted in yellow in the manuscript). Therefore the reference [29] has been removed from the revised version of the article.

Line 82-86: This paragraph is not clear. It is better to put more details about the inert materials (iron oxides) and its effect on the final product.

Response to the reviewer:

This paragraph has been changed according to the reviewer suggestion (line 146-155). The improvement of properties of cement composites with rock dust additives is primarily related to the filler role of rock dust, i.e. mainly the physical mechanism. Therefore the dominate role of filler effect was highlighted. The heterogeneous nucleation of C-S-H on rock dust grains plays the secondary role in shaping the cement paste microstructure.

Table 1: What are the side effects of having high amount of Iron Oxide? Are there any negative impact in long-term?

Response to the reviewer:

This is a scientifically valuable question. First of all, thank you for this question.

As it is known, Portland cement, which is the main binder of concrete production, includes four main compounds. These are tricalcium silicate (3CaO · SiO2), dicalcium silicate (2CaO · SiO2), tricalcium aluminate (3CaO· Al2O3), and a tetra-calcium aluminoferrite (4CaO·Al2O3Fe2O3). This construction material is produced with a closely controlled chemical combination of calcium, silicon, aluminum, iron, and other ingredients. Iron oxide quantity in cement ranges from 0.1% to 7-8%. The main function of iron oxide is to give color to the cement. As it is well known, the cement is a hydraulic binder and reacts chemically in concrete. However, there is no reported knowledge that the iron oxide in cement has a harmful effect on the long-term performance of concrete.

If this situation is considered in terms of concrete aggregates, it only affects long-term performance (such as alkali silicate reaction), if the aggregates have active silica etc. on its surface. The fact that the aggregates are inert and contain some chemically bound oxides does not cause major problems.

Furthermore, it is also known that some aggregates containing high levels of iron oxide are widely used in concrete production as aggregates.

Such aggregates as limonite, magnetite, hematite can be present among aggregates containing high amounts of iron oxide (up to 75-80 %). These aggregates are widely used especially in the production of radiation-shielding concrete. These aggregates have no known major effects in terms of long-term performance of concrete.

Line 158: Do you mean Figure 3, not Figure 1?

Response to the reviewer:

We mean Figure 3, not Figure 1. We are sorry for this mistake.

Line 164: Grammar checks are must. For example, in Line 164 “To analysed” and in Line 171: “To analysis”!

Response to the reviewer:

English revision of the whole article has been performed.

Line 208: Slump target 140 ± 10 mm

Response to the reviewer:

We are sorry for this mistake. It has been changed, i.e. from 1 mm to 10 mm.

Section 3: This section should be sub-divided to discuss different tests and results. In addition, the section presented the discussion before showing results. More scientific elaboration is essential during discussion of results.

Response to the reviewer:

Section 3 (3. Results and Discussion) has been changed according to the reviewer suggestion. More scientific elaboration was added to this part of article. The added sentences were highlighted in yellow in the manuscript. The sentences we have changed the order in the manuscript were marked in blue.

Line 272-273: It worth to explain more about the relation between dispersion and hydration of cement.

Response to the reviewer:

It has been changed according to the reviewer suggestion (line 371-376):

Cement grains, electrostatically charged differently, tend to attract one another in water slurry, resulting in aggregate creation (flocculation). In this case, water is not able to freely penetrate all the space between cement grains and not all cement particles are effectively used in the hydration process. Microfiller introduction causes greater dispersion of cement grains, which contributes to the accelerated hydration of clinker phases, and therefore faster strength increase [12, 33].

Line 295: concrete cure.

Response to the reviewer:

It has been changed, i.e. “care” to “cure” (line 409). We are sorry for this mistake.

Section 4: it looks more like a summary rather than conclusions. It should be more concise.

Response to the reviewer:

The conclusion part of the study has been arranged in such a way to clearly show the results of the study. The new version of conclusions section has been embedded into the manuscript (line 521-572).

Line 408: Authors mentioned “fully justified”. However, there are so many other parameters that are not covered by this article and may have drawbacks on the long-term and durability.

Response to the reviewer:

The conclusion has been completely changed. Therefore this sentence was removed from the article.

Reviewer 2 Report

Important topic, useful outputs  but the language needs minor revision. For example: When you start your sentence with In the paper...… It should be: In this paper,  ….There are several things like that in the manuscript. Please correct all.

Author Response

Dear Reviewer

Thank you for your comments concerning our manuscript entitled “Physical Properties and Microstructure of Concrete with Waste Basalt Powder Addition”. The comments were all valuable and very helpful for revising and improving our paper. We have studied the comments carefully and have made a variety of revisions that we hope will meet with your approval. The revised portions of the manuscript are highlighted in yellow in the manuscript itself.

Is the paper well written? Is the text clear and easy to read?

The language is ok apart from some minor and spelling corrections and the text is easy to read.

Response to the reviewer:

The linguistic corrections of English have been made.

Are the conclusions consistent with the evidence and arguments presented? Do they address the main question posed?

Yes the conclusions are consistent with the evidence and arguments presented in the paper. The conclusions also addressed and answered the main question of the benefit of adding basalt powder.

However, the authors can add some values from their results to strengthen their conclusions.

Response to the reviewer:

Authors added data values from their results to reinforce the conclusions. Section 3 and section 4 has been changed. More scientific elaboration was added to this part of article. The added sentences were highlighted in yellow in the manuscript. The sentences we have changed the order in the manuscript were marked in blue.

Reviewer 3 Report

1) The originality and the scientific value of the subject are very good. An important problem having direct applications is treated.

2) The Abstract in its current form is not sufficient. In particular, it should be supported in a more effective manner by the results obtained during research, because the first part which is read by Journal’s audience is Abstract and thus it should reflect the novelty and perform the main results. 

3) The Introduction Section in its current form is not adequate. In this context, I recommend the authors to further analyze and discuss the results of Refs. [5-8], [9-11], [15-19], [20-22] and [23-27]. In addition, the differences/advantages of the present investigation compared to other literature works should be written out at the end of this Section in a much more thorough and comprehensive manner.

4) The materials, their applications, applied methods and especially the use of the investigated material are explained in detail. The composition, the origin of the material used, dimensions of specimens etc are all mentioned

5) Presentation of the experimental work is very thorough. Process and prerequisites of sample preparation are clearly mentioned. However, the authors are kindly recommended to provide some further technical details about the laboratory equipment that they used to carry out their experiments.

6) The presentation and clarity of results and data are good. However, the discussion of the results is relatively adequate. The authors could give some additional theoretical explanations about Figs. 6, 7, and 8. Moreover, is there any possibility for comparison with theoretical and/or advanced computational methods (FEM, BEM)?

7) Logic and coherence are concrete and the clarity and quality of writing are sound.

8) The "Conclusions" Section performs the findings of this work in an adequate manner. Nonetheless, I invite the authors to add a paragraph on the motives and prospects that this work provides for future research.

Overall, it can be said that the manuscript may be recommended for publication provided that the authors interpret these critical remarks in a constructive manner and revise the manuscript accordingly. 

Author Response

Dear Reviewer

Thank you for your comments concerning our manuscript entitled “Physical Properties and Microstructure of Concrete with Waste Basalt Powder Addition”. The comments were all valuable and very helpful for revising and improving our paper, as well as for helping us to better explain the significance of our research. We have studied the comments carefully and have made a variety of revisions that we hope will meet with your approval. The revised portions of the manuscript are highlighted in yellow in the manuscript itself. In addition, the main revisions to the paper are discussed in our individual responses to your comments that follow below:

2) The Abstract in its current form is not sufficient. In particular, it should be supported in a more effective manner by the results obtained during research, because the first part which is read by Journal’s audience is Abstract and thus it should reflect the novelty and perform the main results.

Response to the reviewer:

Abstract section has been revised. The new version of abstract has been included:
1) the overall purpose of the investigated study; 2) the overall details of the study; 3) major findings as a result of the research. The revised abstract has been embedded into the manuscript (line 11-26 highlighted in yellow in the manuscript).

3) The Introduction Section in its current form is not adequate. In this context, I recommend the authors to further analyze and discuss the results of Refs. [5-8], [9-11], [15-19], [20-22] and [23-27]. In addition, the differences/advantages of the present investigation compared to other literature works should be written out at the end of this Section in a much more thorough and comprehensive manner.

Response to the reviewer:

All references mentioned above by the reviewer have been re-read and detailed in the introduction section. And the advantages of the present investigation have been written in a much more thorough and comprehensive manner (line 56-73, line 114-145). The following paragraph has been also added to the introduction section:

“Despite the growing interest in the use of various rock dust types in concrete production, many problems still remain unexplained. There is no literature related to basalt dust utilization as a partial sand substitute in concrete. The scientific goal of the research presented in the paper was to determine the effect of the basalt dust additive, partially replacing sand, on the properties of concrete. The influence of basalt dust on technological properties of concrete mixtures, compressive strength, water absorption, permeability, and microstructure of hardened concrete with basalt dust additive were analyzed. The results of research on the concrete mechanical properties presented by the authors in this paper are consistent with those of other scientists who analyzed the influence of rock dusts of different mineral origin (lime, marble and granite dusts) on concrete properties. Moreover, the presented analyses concerning the influence of basalt dust on the porosity and microstructure of the cement matrix supplement the knowledge deficiency, especially with regard to basalt dusts used in concrete as a sand substitute.”

5) Presentation of the experimental work is very thorough. Process and prerequisites of sample preparation are clearly mentioned. However, the authors are kindly recommended to provide some further technical details about the laboratory equipment that they used to carry out their experiments.

Response to the reviewer:

The compression tests were performed using a computer-controlled test machine having a capacity of 3000 kN. The hardness value of the compressive test machine's loading heads is 550 HV 30 (HRC 53) which confirms to the EN12390-4. For tests, the loading rate was selected constant as 0.5 MPa/s.

Permeability of concrete was determined with a measuring device designed to determine the depth of water penetration in hardened concrete specimens under pressure. Research was conducted in accordance with the procedure set out in EN 12390-8. The measuring device enables the transfer of water pressure to the test area and its current indications.

SEM analyses were performed by FEI brand Quanta FEG 250 model instrument.

6) The presentation and clarity of results and data are good. However, the discussion of the results is relatively adequate. The authors could give some additional theoretical explanations about Figs. 6, 7, and 8. Moreover, is there any possibility for comparison with theoretical and/or advanced computational methods (FEM, BEM)?

Response to the reviewer:

  1. Some additional explanations about Figures 6, 7, and 8 have been added to the manuscript.
  2. The FEM is used for solving problems of engineering and mathematical models related to the fields of structural analysis, heat transfer, fluid flow, mass transport, and electromagnetic potential. The FEM is a particular numerical method for solving partial differential equations. The Boundary Element Method (BEM) is a numerical computational method of solving linear partial differential equations, including fluid mechanics, acoustics, electromagnetics, fracture mechanics, etc.

As far as we know (we are not sure, though), these approaches are generally preferred by those who do theoretical work in the fields of engineering. Construction materials and concrete technology researchers generally conduct applied laboratory studies and deal with experimental findings by using relative comparisons. If it is allowed and approved by the reviewer, we are not willing to add such analysis, as FEM, BEM, which is not related to our expertise area.

8) The "Conclusions" Section performs the findings of this work in an adequate manner. Nonetheless, I invite the authors to add a paragraph on the motives and prospects that this work provides for future research.

Response to the reviewer:

The following paragraph has been added to the conclusions section:

“This study may be extended with new experimental studies related to concrete durability and steel-concrete bond performance of basalt dust substituted in concrete to better understand the potential of basalt powder utilization in concrete production.”

Round 2

Reviewer 1 Report

Many thanks for the improvements.

Reviewer 3 Report

I read the revised manuscript very carefully and came to the conclusion that the authors made a very significant effort to improve their article so that it now meets the high quality standards of “Materials”. Furthermore, after much consideration, I have to say that I agree with the authors' view on the use of the popular computational methods FEM and BEM. In this framework, I am satisfied with the manuscript in its current form and I recommend it for publication.